# Genome-Wide Identification, Characterization, and Expression Analysis of the *BES1* Family Genes under Abiotic Stresses in *Phoebe bournei*

**DOI:** 10.3390/ijms25053072

**Published:** 2024-03-06

**Authors:** Jingshu Li, Honggang Sun, Yanhui Wang, Dunjin Fan, Qin Zhu, Jiangyonghao Zhang, Kai Zhong, Hao Yang, Weiyin Chang, Shijiang Cao

**Affiliations:** 1College of Forestry, Fujian Agriculture and Forestry University, Fuzhou 350002, China; chuchu7613@163.com (J.L.); fandunjin@foxmail.com (D.F.); 15779224008@163.com (Q.Z.); zjyhssg@163.com (J.Z.); m15803366858@163.com (H.Y.); 2Research Institute of Subtropical Forestry of Chinese Academy of Forestry, Hangzhou 311400, China; honggangsun@caf.ac.cn; 3College of Horticulture, Fujian Agriculture and Forestry University, Fuzhou 350002, China; w19819995339@163.com; 4Laboratory of Virtual Teaching and Research on Forest Therapy Specialty of Taiwan Strait, Fujian Agriculture and Forestry University, Fuzhou 350002, China; 5College of JunCao Science and Ecology, Fujian Agriculture and Forestry University, Fuzhou 350002, China; kaichung2024@163.com

**Keywords:** *Phoebe bournei*, brassinosteroid (BR), *BES1* genes, expression profiling, abiotic stresses

## Abstract

The BRI1 EMS suppressor 1(BES1) transcription factor is a crucial regulator in the signaling pathway of Brassinosteroid (BR) and plays an important role in plant growth and response to abiotic stress. Although the identification and functional validation of *BES1* genes have been extensively explored in various plant species, the understanding of their role in woody plants—particularly the endangered species *Phoebe bournei* (Hemsl.) Yang—remains limited. In this study, we identified nine members of the *BES1* gene family in the genome of *P. bournei*; these nine members were unevenly distributed across four chromosomes. In our further evolutionary analysis of *PbBES1*, we discovered that *PbBES1* can be divided into three subfamilies (Class I, Class II, and Class IV) based on the evolutionary tree constructed with *Arabidopsis thaliana*, *Oryza sativa*, and *Solanum lycopersicum*. Each subfamily contains 2–5 *PbBES1* genes. There were nine pairs of homologous *BES1* genes in the synteny analysis of *PbBES1* and *AtBES1*. Three segmental replication events and one pair of tandem duplication events were present among the *PbBES1* family members. Additionally, we conducted promoter *cis*-acting element analysis and discovered that *PbBES1* contains binding sites for plant growth and development, cell cycle regulation, and response to abiotic stress. *PbBES1.2* is highly expressed in root bark, stem bark, root xylem, and stem xylem. *PbBES1.3* was expressed in five tissues. Moreover, we examined the expression profiles of five representative *PbBES1* genes under heat and drought stress. These experiments preliminarily verified their responsiveness and functional roles in mediating responses to abiotic stress. This study provides important clues to elucidate the functional characteristics of the *BES1* gene family, and at the same time provides new insights and valuable information for the regulation of resistance in *P. bournei*.

## 1. Introduction

Plants often suffer from a series of biotic and abiotic stresses, leading to a decline in yield and quality. Transcription factors (TFs) play a crucial role in plant growth and development, as well as stress response regulation, by activating or inhibiting the transcription of target genes [1]. In plants subjected to biotic and abiotic stresses, TFs can also activate various defense mechanisms [2]. The BRI1 EMS suppressor 1(BES1) transcription factor is a crucial regulator in the signaling pathway of Brassinosteroid (BR), which is a plant steroid hormone engaged in plant growth and development as well as environmental stress responses [3,4,5]; the application of exogenous BR has been shown to promote plant growth under abiotic stress conditions [6]. BR is the sixth type of phytohormone after auxin and was discovered in *Brassica napus*, gibberellins, cytokinin, abscisic acid, and ethylene [7,8]. BR induces a local accumulation of hormones, which triggers signaling for the gradual transformation of cells in the meristematic zone into the elongation zone and promotes cell elongation. The correct balance of BR levels appears to be crucial for normal root growth and development [9,10]. In *A. thaliana*, the deletion mutants of BR exhibit dwarfism, reduced cell elongation, reduced apical dominance, delayed flowering and senescence, and male sterility [11,12]. The signaling pathway of BR can interact with multiple responses and cooperatively regulate plant growth, development, and response to abiotic stresses [7]; this pathway also cross-regulates with other pathways, including the ABA, light, and GA signaling pathways [13,14,15], through the *BES1* family of transcription factors. In response to abiotic stresses, the BR signaling pathway negatively regulates the plant’s drought stress response by crosstalking with drought stress through RD26 [16,17]. The BR signaling pathway also interacts with ABA signaling to enhance plant resistance to abiotic stresses [15].

BZR1/BES1 is a plant-specific transcription factor that can bind to and regulate BR-responsive genes [18,19]. Research indicates that BZR1 and BES1 are two closely related nuclear proteins that share high homology in their amino acid sequences. They bind to DNA through a conserved N-terminal DNA-binding domain [20], and they associate with genes related to BR and regulate their expression [21,22,23]. Both BZR1 and BES1 play positive regulatory roles in the BR signaling pathway [24]. In plant cells, BR binds directly to the extracellular structural domain of Brassinosteroid insensitive 1 (BRI1) and activates its intracellular activating enzyme activity [25]. This promotes the dissociation of BRI1 kinase inhibitor 1 (BKI1) from the cell membrane, allowing it to bind to BRI1-associated kinase 1 (BAK1) and to signal BR into the cell [26]. In the cytoplasm, BR-signaling kinase 1 (BSK1) through a phosphorylation/dephosphorylation cascade, *BSK*, which in turn activates BRI1-suppressor 1 (BSU1), which dephosphorylates and degrades Brassinosteroid-insensitive 2 (BIN2) [27]. This rescues the inhibition of *BES1* by *BIN2* [18,28]. *BES1* is dephosphorylated by protein phosphatase 2A (PP2A) and accumulates in the nucleus, where it acts on a large number of target genes downstream of *BES1* [5,29]. For example, *BES1* binds to *ACO1* to increase ethylene production and promotes gravitropic responses in *A. thaliana* roots [30]. BES1 possesses an unusual bHLH structural domain, which is the largest family of transcription factors in the model plant *A. thaliana* [31,32]. This domain can regulate the expression of thousands of downstream genes in conjunction with the E-box of the promoter region of the target genes (CANNTG) and the oleoresin lactone response element (CGTGT/CG) [33,34]. Furthermore, the *BES1* gene family interacts with *PIF4*, *WRKY46*, *WRKY54*, and *WRKY70* to co-regulate plant cell elongation and plant stress tolerance [35,36]. Mecchia et al. demonstrated that the sporophyte geophyte *BES1* plays a crucial role in controlling cell proliferation and differentiation [37]. In a study of the *BES1* gene family in the angiosperm *Populus trichocarpa*, it was found that *BES1* enhances the drought stress tolerance and scavenging of reactive oxygen species in plants [38].

Recently, it has been discovered that *A. thaliana* [23] has 8 members of the BES1 transcription factor family, rapeseed [39] has 28, cotton [40] has 22, corn [41] has 11, tomato [42] has 9, and cabbage [43] has 15. Previous research has demonstrated the involvement of BES1 transcription factors in plant growth, development, and stress responses, such as cell growth [44], pollen development [45], plant immune signaling [46], and resistance physiology [23]. Studies indicate that BES1 is involved in the formation of the primary periderm; it inhibits the genes AIB3 and AIB5, which are insensitive to abscisic acid (ABA), attenuates ABA signaling during seedling development, accelerates flowering in the reproductive stage, and regulates elongation of the hypocotyl [47]. In cotton research, the *GhBES1* gene exhibits functional diversity, affecting fiber growth and consequently influencing cotton plant structure [41]. Additionally, research shows that the expression levels of most *BES1* genes change significantly under hormone stimulation, indicating that BES1 transcription factors mediate plant responses to hormonal stress. The cotton variety ‘Xinluzao 17’ rapidly responds to drought stress under the regulation of BES1 transcription factors [48]. Furthermore, studies suggest that the majority of tomato *BES1* genes are significantly up-regulated 6 and 24 h after salt treatment, emphasizing the crucial role of the BES1 transcription factor family in salt tolerance in tomatoes [42].

*Phoebe bournei* (Hemsl.) Yang, which belongs to the Lauraceae family and Phoebe genus, stands as a renowned arboreal species in the horticultural field. This species holds a prominent position among the flora within subtropical evergreen broad-leaved forests and has substantial economic worth and ecological relevance [23]. Due to commercial and development needs, *P. bournei* has been heavily cut down and destroyed in recent years, leading to a gradual decrease in its population. Its slow growth has resulted in its classification as an endangered plant. *P. bournei* grows in monsoon climates with abundant precipitation. However, its growth is affected by droughts and prolonged summers [49]. Studies have shown that global warming, caused by the rising global carbon dioxide concentrations due to the combustion of industrial fuels, significantly affects the incidence of forest fires and the efficiency of photosynthesis in forest trees [50,51]. Additionally, global warming contributes to the dryness of the climate [49]. Drought has a significant impact on the dry weight of stems and roots, as well as chlorophyll synthesis and photosynthesis in *P. bournei* [52]. Abiotic stresses can also affect chlorophyll biosynthesis, leading to reduced photosynthesis [53]. Therefore, it is highly desirable to improve the plants’ ability to respond to abiotic stresses.

This study analyzed the physicochemical properties, evolutionary relationships, gene structure, conserved structural domains, and intraspecies and interspecies covariance of the *BES1* gene family of *P. bournei*. It also investigated and determined the expression of the *PbBES1* family of genes under different abiotic stresses, providing novel insights and information for future research on the selection and regulation of stress tolerance. Furthermore, it offers valuable insights and information for the further understanding of the functional characteristics of the *BES1* gene family.

## 2. Results

### 2.1. Analysis of Physicochemical Properties of the BES1 Gene Family in Phoebe bournei

Predictive analysis of the physicochemical properties of the amino acid sequences of nine members of the *PbBES1* family was conducted; these nine members were named *PbBES1.1*~*PbBES1.9* (Table 1). They spanned a wide range of amino acids, from 134 (*PbBES1.8*) to 696 (*PbBES1.6*). The protein had a molecular weight ranging from 15,094.98 kDa (*PbBES1.8*) ~ 78,314.37 kDa (*PbBES1.6*), and the theoretical isoelectric point ranged from 5.59 (*PbBES1.9*) to 10.21 (*PbBES1.3*), with a mean value of 7.44. In addition, four of the genes had larger isoelectric points; the differences in isoelectric points indicated that *PbBES1s* function in different microenvironments. Additionally, all nine *PbBES1s* had negative hydrophilicity, and the lipid solubility index was located between 50.56 (*PbBES1.2*) and 75.66 (*PbBES1.9*), making them hydrophilic proteins. Four genes had a lipid solubility index greater than seventy and were thermally stable. The proteins with instability coefficients of less than 40 are generally considered to be stable proteins, and the proteins with coefficients greater than 40 are considered to be unstable proteins. Among the nine members of the PbBES1 family, only PbBES1.9 had an instability coefficient lower than 40, indicating that it belongs to the stable proteins. The remaining members of the PbBES1 family were unstable proteins.

### 2.2. Protein Evolution and Collinearity Analysis of PbBES1 Genes

Phylogenetic trees display the affinities and evolutionary relationships between individual species and genes. To investigate the evolutionary relationship between the *PbBES1* gene family and other species, we performed multiple sequence comparisons and constructed maximum likelihood phylogenetic trees with the *PbBES1s* using three representative plants: *Arabidopsis thaliana* (14), *Solanum lycopersicum* (9), and the monocotyledonous plant *Oryza sativa* (7) (Figure 1). The distribution of *PbBES1* genes among the different fractions was uneven, with Class I having a maximum of five *PbBES1*s. Class II contained two *PbBES1*s, followed by two *PbBES1*s in Class IV. The phylogenetic tree revealed that *PbBES1.6* was closely related to *PbBES1.9*, while *PbBES1.1* was closely related to *PbBES1.2* and *PbBES1.3* was closely related to *PbBES1.4.* Notably, there were no *PbBES1* genes in Class III, and the *BES1* genes were only distributed in dicotyledonous *A. thaliana* and *S*. *lycopersicum*.

The evolutionary tree analysis revealed that the *BES1* family members were interspersed in four plants, but Class III contained only the *BES1* genes of *A. thaliana* and *S. lycopersicum*. This suggests that the *BES1* of *P. bournei* may have amplified after the monocotyledonous divergence, which was more conserved. Martin analyzed the evolutionary tree of *BES1* and suggested that it first appeared in bryophytes and may have developed after phytoplankton [37]. Furthermore, the evolutionary origins of *PbBES1.6* and *PbBES1.9* are homologous to those of *PbBES1.5* from the consensus branch.

The collinearity model provides an historical insight into the genome and enables downstream analyses [54]. Following the analysis of the *PbBES1* gene family (Figure 2), its intraspecies covariance showed that the 9 *PbBES1* genes were mainly located on 4 of the 12 chromosomes of *P. bournei.* The chromosome Chr01 contained three *PbBES1*s and was the chromosome with the highest number of *PbBES1* genes, with two genes distributed on each of the remaining three chromosomes. The *PbBES1* genes had a total of four replication events: *PbBES1.3* on Chr01 and *PbBES1.4* were tandem replication events; *PbBES1.8*/*PbBES1.9/PbBES1.6* and *PbBES1.2/PbBES1.1* were fragment replication events.

The abundance of duplicate genes in plant genomes contributes to the evolution of new functions [38,55]. Tandem duplication events are genes that contain two or more copies within a region smaller than 200 kb. Fragment duplication events, on the other hand, are homologous gene pairs found on different chromosomes [56].

To gain a deeper understanding of the evolutionary mechanism of *P. bournei*, interspecific collinearity analyses were performed between *P. bournei* and the three representative species used to construct the evolutionary tree (Figure 3). Four homologous gene pairs were identified between *PbBES1* and *OsBES1*. Additionally, eight homologous gene pairs were identified between *PbBES1* and *SlBES1,* and *PbBES1* had nine homologous gene pairs with *AtBES1.* Analyzing the collinearity between species can provide an insight into the timing of gene family amplification.

### 2.3. Analysis of Gene Structure and Conserved Motifs in Phoebe bournei

The analysis of the 9 identified *PbBES1* family members revealed that they had 10 conserved motifs (Figure 4A). The members of the same subfamily shared the same motif composition and arrangement order. All the genes contained motif 1 and motif 8, indicating that these are the most conserved motifs. PbBES1.7 and PbBES1.8 only contained motif 8 and motif1, while PbBES1.1, PbBES1.2, PbBES1.3, and PbBES1.4 contained motif8, motif1, and motif2. This suggests that they may have similar functions. The differences in conserved motifs among the members of the *PbBES1* family suggest that gene loss or deletion may have occurred during the evolutionary process. On the conservative structural domains (Figure 4B), PbBES1.9 and PbBES1.8 had two conserved structural domains, AmyAc family and BES1. The remaining family members possessed the BES1 structural domain, suggesting a higher degree of conservation.

The analysis of the gene structure revealed that there were nine genes with exons ranging from two to nine and introns ranging from one to nine. The number of exons and introns in *PbBES1.9*, *PbBES1.6*, and *PbBES1.5* was nine. The numbers of exons and introns in *PbBES1.8*, *PbBES1.7*, *PbBES1.4*, *PbBES1.3*, *PbBES1.2*, and *PbBES1.1* were two and nine, respectively. Furthermore, it is worth noting that *PbBES1.8*, *PbBES1.4*, and *PbBES1.3* lacked non-coding regions. These regions play a crucial role in regulating the gene expression of mRNA stability.

The protein conserved structural domains are highly conserved, and protein sequence similarity can provide valuable clues to evolutionary processes [57]. To further investigate the evolution and function of the *PbBES1* gene family, we performed multiple sequence comparisons of its conserved structural protein domains (Figure 5). With reference to previous studies [19,58], comparisons were made with the structural domains of the AtBES1 protein. The conserved structural domains of the PbBES1 family proteins were found to have atypical basic helix–loop–helix (bHLH) structural domains and highly conserved amino-terminal structural domains (N) and *BIN2* phosphorylation binding sites (P). This suggests that the *PbBES1* gene family is more conserved in the evolutionary processes.

### 2.4. Cis-Acting Analysis and Structure of the PbBES1 Promoter

To enhance comprehension of the function of the *PbBES1* genes, we predicted the *cis*-acting elements in their promoter regions (Figure 6). The results revealed that the *PbBES*1 gene family contained a total of 15 *cis*-acting elements that are related to environmental stress and hormone response. Each *cis*-element represented a distinct function, with the highest number of homeostatic components being related to light response; there were 34 of these components, and they were contained in the promoter and enhancer regions. All the members, except *PbBES1.6* and *PbBES1.7*, contained light-responsive *cis*-acting elements. This suggests that the *PbBES1* gene family may play a role in regulating plant light response. The *PbBES1.7* genes had the highest number of *cis*-acting elements, with 37 each, indicating their potential involvement in transcriptional regulation. All the genes had phytohormone response elements, and *PbBES1.2*, *PbBES1.5*, *PbBES1.6*, *PbBES1.7*, and *PbBES1.9* also had MYB transcription factor binding sites that were involved in drought induction. The involvement of MYB transcription factors in various abiotic stress responses, including drought stress [59], suggests that the *PbBES1* gene family may also regulate plant resistance to drought stress.

Further analysis of the *cis*-acting progenitor of the *PbBES1* promoter region (Figure 7) was conducted. In the *PbBES1* gene family, there were 36 *cis*-elements related to stress; *PbBES1.7* contained up to 7 stress-responsive *cis*-elements and 52 *cis*-elements related to growth and development; *PbBES1.7* contained up to 13 growth-responsive *cis*-elements and 20 hormone-responsive *cis*-elements; *PbBES1.1*, *PbBES1.2*, and *PbBES1.9* contained 4 hormone-responsive *cis*-elements each and 34 *cis*-elements related to light response; and *PbBES1.6* was the most numerous, with 6 light-responsive *cis*-elements. The *PbBES1* gene family contained hormone-responsive homeotic elements, including abscisic acid-responsive elements and growth hormone-responsive elements. This suggests a potential link between the *PbBES1* gene family and the abscisic acid and growth hormone signaling pathways, which is consistent with previous research [13,60]. Among the nine members of the *PbBES1* family, *PbBES1.7* contained the most stress-corresponding and hormone-responsive homeotic elements. This suggests that *PbBES1.7* may play a major role in the family and that it warrants further study.

### 2.5. Heat Map of PbBES1 Gene Expression in Different Tissues

To enhance comprehension of *PbBES1*′s role in plant growth and development, we analyzed the expression of the *PbBES1* gene family in various organs (Figure 8). We classified the nine genes into three subfamilies. With the exception of *PbBES1.7*, which was not expressed in the leaves, root bark, or stem bark, all the genes were expressed to varying degrees in all five organs. The expression of the *PbBES1.5* gene was highest in the leaves, indicating its involvement in the growth and development of leaf organs. *PbBES1.2*, a separate subfamily, had the highest expression in the root xylem, stem xylem, root bark, and stem bark. Its expression was higher in the root bark than in the other organs, suggesting a strong correlation between *PbBES1.2* and root growth and development. The expression of each gene in the third subfamily was low in all five organs, and *PbBES1.7* only had low expression in the root xylem and stem xylem. This suggests that *PbBES1.7* may be involved in plant growth and development with fewer corresponding functions. The genes in the third subfamily may play other roles.

### 2.6. Abiotic Stress Experiments on the BES1 Gene Family of Phoebe bournei

To further investigate the response of the *BES1* gene family of *P. bournei* to abiotic stresses, we subjected its leaves to high-temperature and drought stress treatments. The leaves of *P. bournei* were collected at 6 h, 12 h, and 24 h after treatment using qRT-PCR to compare their gene expressions with that of leaves untreated with heat and drought under the same culture conditions (Figure 6). We observed changes in the up- and down-regulation of the different genes. The experimental results indicate that the expression of *PbBES1* in the leaves was decreased to varying degrees after 6h under high-temperature stress treatment (Figure 9a). The *PbBES1.3*, *PbBES1.4*, and *PbBES1.5* expression was significantly decreased, with *PbBES1.5* being approximately one-third of the control (ck). The expression of *PbBES1.1*, *PbBES1.2*, and *PbBES1.4* was significantly increased after 12 h of high-temperature stress. Among them, *PbBES1.2*, which was highly expressed in the root bark, showed a 12-fold increase compared to the control group. However, after 24 h of treatment, the expression of all the genes decreased, except for *PbBES1.1*; *PbBES1.3* was only 1/14 of the control.

Under drought stress treatment (Figure 9b), all the genes except *PbBES1.1* were up-regulated after 6 h, and *PbBES1.3* had an expression that was five times higher than that of the control group. After 12 h, all the genes except *PbBES1.3* decreased in expression. After 24 h, the expression of the *PbBES1* family genes was suppressed, and the expression of *PbBES1.3* tended to be absent. This result indicates that the expression of *PbBES1.3* was significantly altered after 6 h of exposure to both drought and high-temperature treatments. This suggests that *PbBES1.3* may play a more significant role in the response to abiotic stresses.

## 3. Discussion

BR signaling enters cells through membrane receptor BRI1 and acts on a series of transcription factors to finally activate BES1, which binds to a large number of downstream target genes and regulates plant growth and development. According to the analysis, there were nine BES1 genes (BES1.1~1.9) in *P. bournei*. The nine *BES1* genes were identified through bioinformatics; this number of genes was lower than that in *P. trichocarpa* (14) [38] and higher than in rice (6) [61]; it was similar to the number in grape (8) [62], and PbBES1s share the same number of genes and similar amino acid counts with tomato BES1s [63]. The physicochemical properties of PbBES1 showed that PbBES1 was an unstable hydrophilic protein that could adapt to the slightly acidic and slightly alkaline environment. The evolutionary tree analysis revealed that the *BES1* family members were interspersed in four plants, but Class III contained only the *BES1* genes of *A. thaliana* and *S*. *lycopersicum*. This suggests that the *BES1* of *P. bournei* may have been amplified after the monocotyledonous divergence, which was more conserved. Martin analyzed the evolutionary tree of *BES1* and suggested that it first appeared in bryophytes and may have developed after phytoplankton [37]. Furthermore, the evolutionary origins of *PbBES1.6* and *PbBES1.9* are homologous to those of *PbBES1.5* from the consensus branch. The expansion of the *PbBES1* gene family mainly relies on segmental duplication events, which is the primary mode of expansion for this gene family. Similarly, in wheat, the expansion of the *BES1* gene number also occurs through segmental duplication, while in gramineous plants, it occurs mainly through segmental duplication [64]. In addition, the collinearity analysis showed that *PbBES1* was more closely related to dicots. Furthermore, introns play a crucial role in the evolution of various plant species [61]. The members of the *PbBES1* family have different exon and intron positions, and they can be divided into two main categories based on their number, indicating changes in the structure of the *PbBES1* gene family members. The variations in the gene structure and conserved protein domains of the *PbBES1* family members may imply differences in their gene functions. The *PbBES1* gene family is highly conserved. PbBES1.7 and PbBES1.8 have fewer motifs compared to the other three genes in the same subfamily. Additionally, their conserved structural domains and gene structures differ, suggesting that *PbBES1.7* and *PbBES1.8* may have been lost or deleted during evolution and that their structures may have changed accordingly. *PbBES1.5*, *PbBES1.6*, and *PbBES1.9* exhibit similar conserved motifs and gene structures; this is supported by the results of the phylogenetic analyses (Figure 1). In contrast to the *PbBES1* gene family, the grape *BES1* [62] gene family contains at least two exons in total, but *VvBES1–3* contains only one exon and no introns. In an analysis that made a comparison with the wheat and foxtail millet *BES1/BZR1* gene family [65], it was shown that *P. bournei* contained fewer homeotic-acting elements (15), but the wheat and foxtail millet *BES1/BZR1* families contained 66 homeotic response elements. This difference may be due to the fact that *P. bournei* has fewer *BES1* genes. Additionally, all three species contained phytohormone response elements, light response elements, and stress response-related elements. An analysis of BES1 gene transcription factor expression during citrus senescence in a study by Izadi et al. [66] also noted the presence of high-temperature stress-related genes among the targets of the CsBES1 transcription factor.

The gene expression analysis showed that the expression of *PbBES1.1~1.5* was down-regulated 24 h after drought treatment. In contrast to the *PbBES1* gene expression results, the expression analysis of *BES1* in *P*. *trichocarpa* showed that most of the *PtrBES1* expression was up-regulated under drought stress for 24 h. This suggests that it plays a positive role in the drought stress response in *P*. *trichocarpa.* This may be due to the influence of external factors such as the environment during the evolutionary process. In a study by Mahesh et al. [67], the aim was to determine whether BR could promote the growth of *Raphanus sativus* seedlings under drought stress by regulating the increase in nucleic acid and soluble protein content and reducing ribonuclease activity; the plants alleviated drought stress by increasing the activities of superoxide dismutase, catalase, and ascorbate peroxidase, but the activities of superoxide dismutase, catalase, and ascorbate catalase in *R*. *sativus* seedlings under drought treatment need to be further studied [14].

BR mitigates high-temperature stress by enhancing enzymatic and non-enzymatic antioxidant defense and glyoxalase systems [68,69]. EBR, which is a type of BR, significantly improved the survival rate of *A. thaliana* seedlings when exposed to 43 °C for a period of time [70]. EBR treatment was applied to *Ficus concinna* seedlings at 40 °C, and the reduced glutathione (GSH) and the ratio of reduced glutathione to oxidative glutathione (GSH/GSSG) of Ficus melisa were significantly increased. EBR treatment reduced 40 °C-induced increases in O^2−^, H_2_O_2_, and MG (Methylglyoxal) levels, and this process was likely associated with a decrease in lipid peroxidation. EBR attenuated the 40 °C-induced oxidative stress by simultaneously increasing non-enzymatic and enzymatic antioxidant responses, as well as MG detoxification systems [67]. Based on the qRT-PCR results, significant differences in the gene expression of the *PbBES1* gene environments were observed under high-temperature conditions. Martins et al. demonstrated [71] that BR regulates thousands of genes in roots that are regulated by high temperatures. In addition, in a model that simulates the BR signaling pathway and downstream target genes of *BES1* (Figure 10), we found that BES1 was able to bind to the PIF4, which is involved in auxin synthesis, thereby promoting the synthesis of IAA [36]. Therefore, we speculate that *PbBES1* promotes auxin synthesis by activating *PIF4*, thereby alleviating the impact of auxin synthesis and transport at high temperatures and regulating plant growth; however, this needs to be further verified. The expression levels of *PbBES1.1~1.5* under high-temperature stress showed that only *PbBES1.1* was up-regulated after 24 h of high-temperature treatment, and the expression of the other four genes was down-regulated; thus, *PbBES1.1* may be the key gene of the high-temperature stress. Further research, such as that which considers the knockout of *PbBES1.1*, may provide a more accurate understanding of the function of the gene.

## 4. Materials and Methods

### 4.1. Genomic Data

The genomic data and annotation information of *P. bournei* were downloaded from the sequence library of the Chinese National Genebank Database (CNSA), search number CNP0002030 [73]. The genomic data and annotation information for the *A. thaliana*, tomato, and rice were obtained from EnsemblPlents (https://plants.ensembl.org/index.html, accessed on 24 June 2023) and Phytozome v13 (https://phytozome-next.jgi.doe.gov, accessed on 24 June 2023), respectively. The RNA-seq data of the different tissues of *P. bournei* were downloaded from BioProject (https://www.ncbi.nlm.nih.gov/bioproject/, accessed on 24 June 2023) under the accession number PRJNA628065.

### 4.2. Plant Material Sources and Abiotic Stress Treatment

#### 4.2.1. Plant Material Sources

The plant material was obtained from 1-year-old *P. bournei* seedlings that were cultured in an artificial climatic chamber with various stress treatments. The *P. bournei* samples were collected and stored in liquid nitrogen at −80 °C for RNA extraction.

#### 4.2.2. Drought Stress Treatment

Six seedlings from the control group were washed root and soaked in distilled water. Meanwhile, the treatment group was soaked in a nutrient solution containing 10% Polyethylene glycol 6000 (PEG6000) [38,67] and incubated in an artificial climate chamber at a temperature of 25 °C and a humidity of 75%. The treatment group was sampled at 6 h, 12 h, and 24 h time periods with 6 plants in each time period, and the control group was sampled at 0 h.

#### 4.2.3. Temperature Handling

For the high-temperature treatment, 6 plants from the control group were kept at room temperature, while the treatment group was incubated at 40 °C in a warm box. The treatment group was sampled at 6 h, 12 h, and 24 h time periods, with 6 plants sampled at each time point. The control group was sampled at 0 h.

### 4.3. Identification and Physical and Chemical Property Analysis

The protein sequences of the *AtBES1* gene family were obtained from plantTFDB (http://planttfdb.gao-lab.org/ (accessed on 24 June 2023)), and the members of the BES1 family of *P. bournei* were initially screened by removing duplicates with local BLASTp against the *P. bournei* protein sequences. The BES1 structural domain number PF05687 was then obtained by searching the pfam database (http://pfam.xfam.org/ (accessed on 24 June 2023)), and the initial screened protein sequences were further examined using the default parameters of HMMER-3.2.1 (http://hmmer.org/download.html (accessed on 24 June 2023)) with an e-value of <105 to obtain the final *P. bournei* BES1 family members. Then, their physicochemical properties were analyzed using ExPASy (https://web.expasy.org/prot-param/ (accessed on 24 June 2023)).

### 4.4. Phylogenetic Tree Construction

*P. bournei*, *A. thaliana*, tomato, and rice were compared using the Muscle program of MEGA(version 7.0.26 (7170509-x86_64)), and the maximum likelihood trees were constructed using the default settings (bootstrap:1000). iTOL (https://itol.embl.de/, accessed on 26 June 2023) was used to improve and beautify the phylogenetic trees.

### 4.5. Chromosome Distribution and Covariance Analysis

The positional information of the *PbBES1* gene was extracted from the genome (FASTA) file and annotation (GFF) file of *P. bournei* using TBtools(version 1.108). The covariate relationships between *A. thaliana* and *P. bournei*, rice and *P. bournei*, and tomato and *P. bournei* were analyzed using MCScanX (https://github.com/wyp1125/MCScanX/ (accessed on 18 July 2023)) software, respectively, and later visualized using TBtools (version 1.108).

### 4.6. Gene Family Conserved Motifs, Gene Structure Analysis

The PbBES1 protein sequence was characterized using the online software MEME (http://meme-suite.org/ (accessed on 24 June 2023)) with a motif number prediction of 10. A Batch CD-search search with default parameters (https://www.ncbi.nlm.nih.gov/Structure/index.shtml (accessed on 24 June 2023)) was used to detect the conserved structural domains of the PbBES1 proteins.

### 4.7. Multiple Sequence Comparison and Promoter Cis-Element Analysis of the PbBES1 Gene

Jalview software (v2.11. 3.0) was used to perform the multiple sequence comparison of the 9 *PbBES1* genes. To explore the *cis*-acting elements in the sequences, we analyzed the *cis*-regulatory elements in the promoter region of the *PbBES1* gene using the online software PlantCARE (https://bioinformatics.psb.ugent.be/webtools/plantcare/html/ (accessed on 25 June 2023)). After screening and classification, the data were visualized using TBtools (version 1.108) software.

### 4.8. The Expression Profiles of PbBES1 Genes

The expression data for the *BES1* genes in *P. bournei* across the various tissues were obtained from the BioProject database (Appendix A). TBtools (version 1.108) software was employed to analyze these expression data and to construct a gene expression heat map, offering a visual representation of the patterns and levels of gene expression.

### 4.9. RNA Extraction and qRT-PCR Analysis

An RNA extraction kit (HiPure Plant RNA Mini Kit from Magen, Shanghai, China) was used to extract RNA, while cDNA was synthesized using the Prime Script RT reagent Kit (Perfect Real Time from Takara, Dalian, China). TBtools (version 1.108) software was used to design specific primers in the non-conserved region of the target gene (Appendix A); the primers were synthesized by Fuzhou Qingbaiwang Biotechnology Company. Real-time fluorescence quantitative analysis was used with cDNA template (1 µL), cDNA template SYBR Premix Ex TaqTM II (10 µL), specific primers (2 µL), and a ddH2O reaction program (7 µL): 95 °C for 30 s; 95 °C for 5 s; 60 °C for 30 s; 95 °C for 5 s; 60 °C for 60 s; and 50 °C for 30 s, with 40 cycles in total. The internal reference gene was PbEF1α (GenBank No. KX682032) [74]. The expression level of the target gene was calculated using the 2^−∆∆Ct^ method, and the quantitative data were analyzed via a test using GraphPad Prism 9.5 software. Finally, GraphPad Prism 9.5 was used to construct the graphs.

## 5. Conclusions

In conclusion, the *PbBES1* genes play a crucial role in the growth of *P. bournei*, which has great commercial value and development potential. This study identified nine *BES1* genes in *P. bournei* and performed a comprehensive bioinformatics analysis. The phylogenetic analysis classified the nine genes into three subgroups. The evolutionary analysis revealed that *PbBES1* is highly conserved and primarily amplified through fragment duplication. *PbBES1* contains numerous *cis*-acting elements related to environmental stress and hormonal responses, particularly with regard to high-temperature and drought stress. These findings are also supported by the qRT-PCR results. After the stress treatments, *PbBES1.1–1.5* exhibited varying degrees of expression effects, with *PbBES1.1* and *PbBES1.3* showing more significant responses to both stresses. They were significantly expressed after 12 h of high-temperature stress. The study offers a supplementary investigation of the *BES1* gene family in *P. bournei* and provides novel insights and information for future research on the selection and regulation of stress tolerance. Furthermore, it offers valuable insights and information for the further understanding of the functional characteristics of the *BES1* gene family.

## Figures and Tables

**Figure 1 ijms-25-03072-f001:**
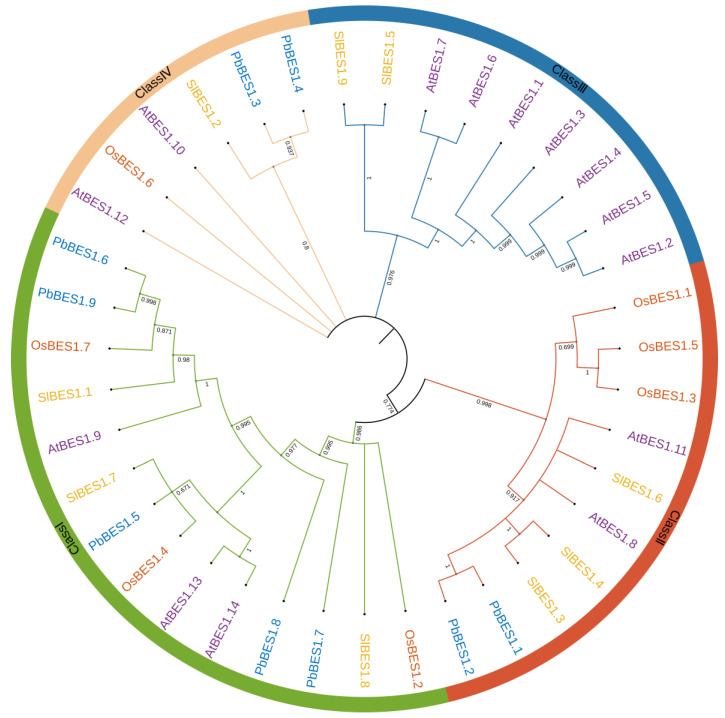
The phylogenetic tree of BES1 proteins in *Phoebe bournei*, *Arabidopsis thaliana*, *Solanum lycopersicum*, and *Oryza sativa*. Classes I–IV referred to the phylogenetic tree clusters.

**Figure 2 ijms-25-03072-f002:**
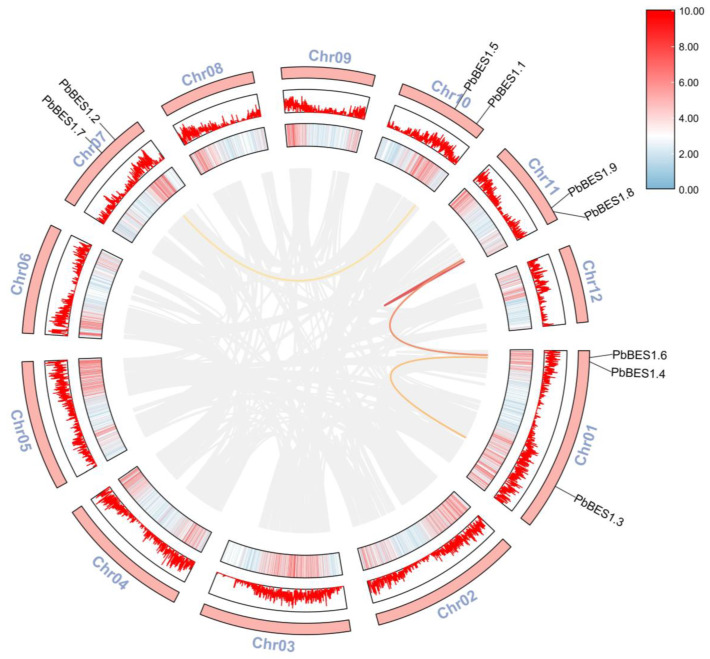
Chromosomal distribution and inter-chromosomal relationship of *PbBES1* genes. The outer red bar indicates the *Phoebe bournei* chromosome, the middle red box indicates the corresponding GC relationship of each chromosome, and the colored line represents the covariance of *PbBES1*; the gray line indicates the covariance of the whole *Phoebe bournei* genes.

**Figure 3 ijms-25-03072-f003:**
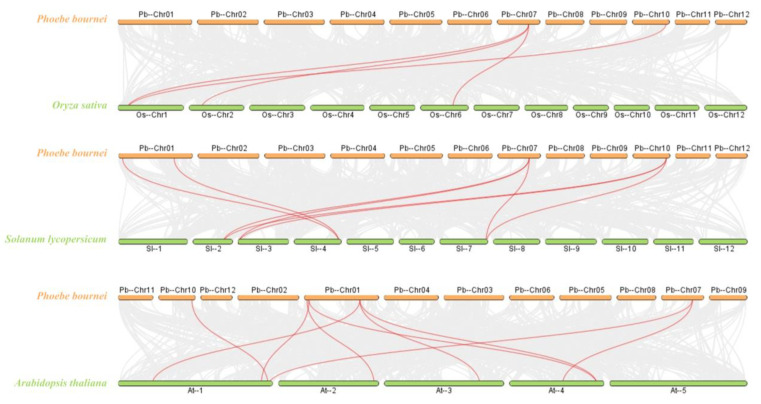
Collinearity analysis plot of *Phoebe bournei* with *Oryza sativa*, *Solanum lycopersicum*, and *Arabidopsis thaliana*. The red line represents the collinearity comparison of the *BES1* gene family, and the gray portion represents the collinearity comparison of the other gene families.

**Figure 4 ijms-25-03072-f004:**
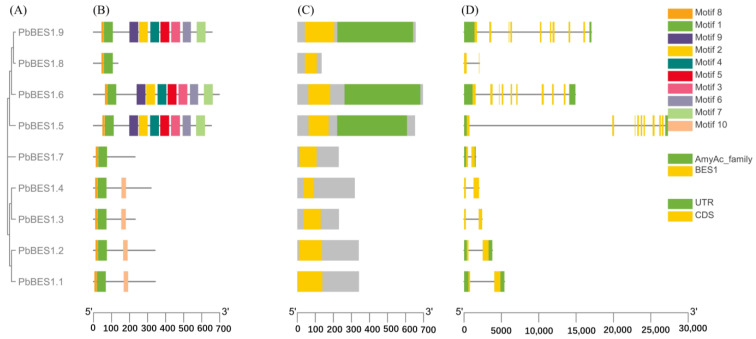
Schematic diagram of the conserved motif of the *PbBES1* gene. (**A**) Phylogenetic tree of 9 PbBES1 proteins. (**B**) Different colors correspond to different types of motifs with the numbers 1–10. (**C**) Indicates a conserved structural domain of PbBES1. (**D**) Exon/intron structure of the *PbBES1* gene. The exon is represented by the yellow box, while the intron is represented by the black line. The green box indicates the *PbBES1* gene’s UTR region.

**Figure 5 ijms-25-03072-f005:**
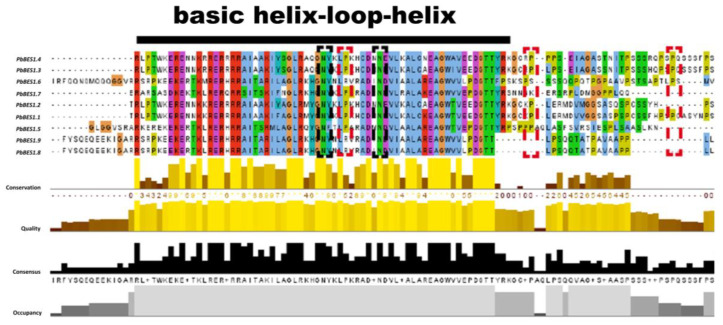
Multiple sequence comparison of PbBES1 protein using Jalview software (v2.11. 3.0). Different amino acids are labeled with different colors, and the possible functional sites or elements are encircled by a box. Black line shows portion of BHLH conserved sequence; black dashed box shows conserved amino-conserved structural domains; red dashed box shows conserved BIN2 phosphorylation binding site.

**Figure 6 ijms-25-03072-f006:**
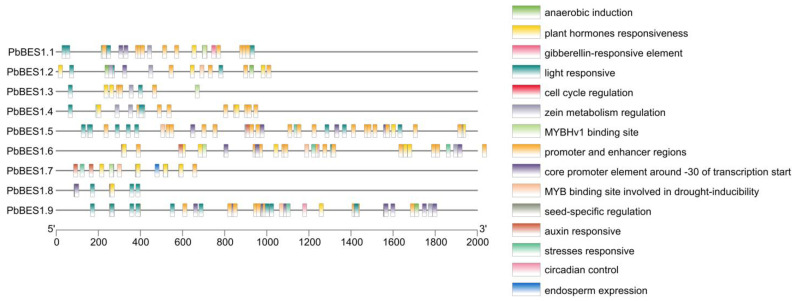
Analysis of *cis*-acting elements of the *PbBES1* gene family, with different colored squares representing different *cis*-elements; the scale bar at the bottom indicates the position of different cis-elements in the promoter region.

**Figure 7 ijms-25-03072-f007:**
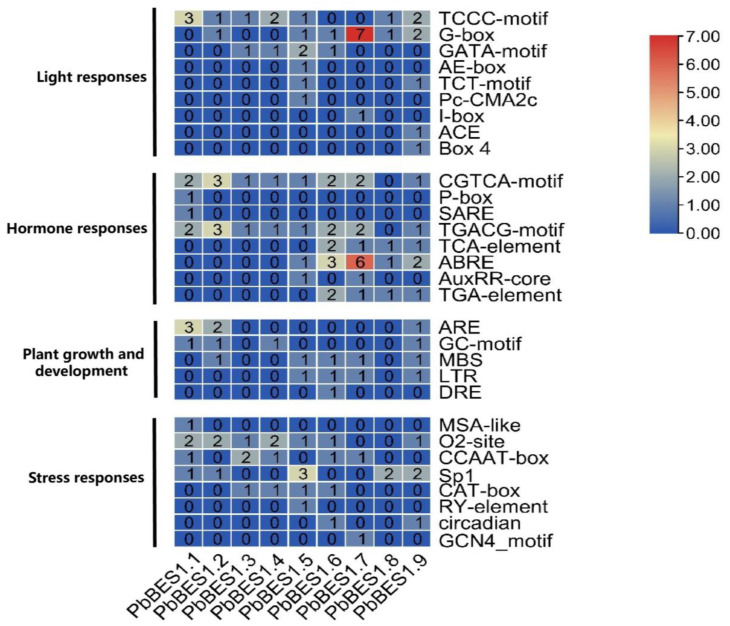
The numbers of the 30 *cis*-elements of the 9 *PbBES1* genes.

**Figure 8 ijms-25-03072-f008:**
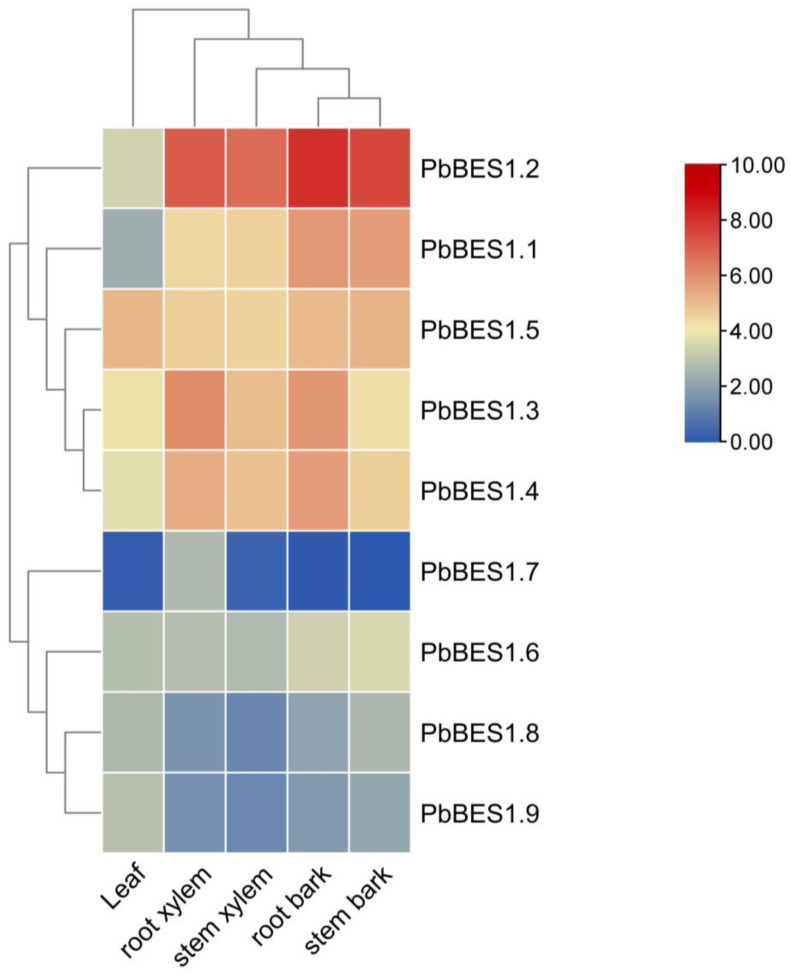
Heat map of *PbBES1* gene family expression in leaves, root xylem, stem xylem, root bark, and stem bark; different colors represent different levels of expression, with blue representing the lowest expression and red representing the highest.

**Figure 9 ijms-25-03072-f009:**
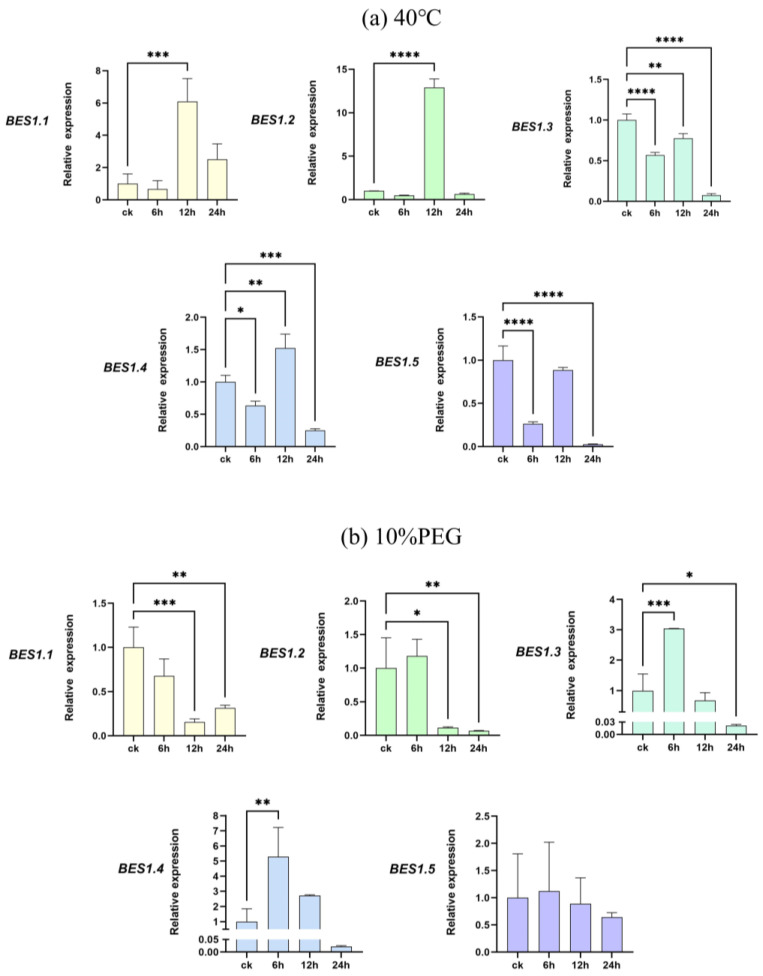
The expression profiles of five representative *PbBES1* genes in response to high-temperature (**a**) and drought (**b**) stresses. The relative expression levels of *PbBES1* genes in response to abiotic stresses, assessed using RT-qPCR. The error bars indicate the standard deviations of the three independent RT-qPCR biological replicates. (X: process time; Y: relative expression) * represents a significant difference relative to the 0 h group (* *p* < 0.05, ** *p* < 0.01, *** *p* < 0.0005, **** *p* < 0.0001).

**Figure 10 ijms-25-03072-f010:**
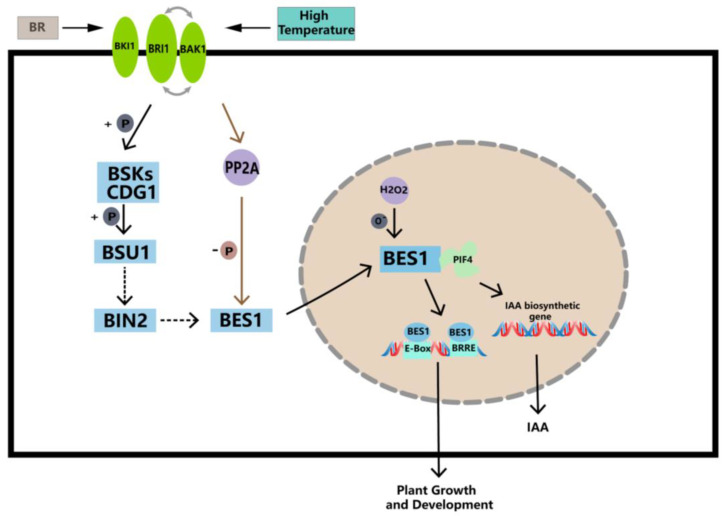
Working model of *BES1* transcription factor under high-temperature stresses. Activated *BES1* interacts with *PIF4*, which promotes the expression of IAA synthesis genes and increases IAA content in high-temperature environments, thereby increasing the expression of *BES1* genes and enhancing the ability of plants to cope with high temperatures (modified from Li et al. [47]; Choudhary et al. [72]).

**Table 1 ijms-25-03072-t001:** Physicochemical characterization of 9 *PbBES1* genes and their encoded proteins.

Sequence ID	Number of Amino Acid	Molecular Weight (kDa)	Theoretical pI	Instability Index	Aliphatic Index	Grand Average of Hydropathicity (GRAVY)
*PbBES1.1*	341	36,251.28	8.33	58.07	55.87	−0.568
*PbBES1.2*	340	36,470.59	7.14	59.20	50.56	−0.598
*PbBES1.3*	230	24,733.89	10.21	92.65	61.22	−0.632
*PbBES1.4*	318	33,920.70	9.14	75.55	56.89	−0.629
*PbBES1.5*	652	73,573.82	5.81	43.65	72.42	−0.485
*PbBES1.6*	696	78,314.37	5.65	43.90	71.47	−0.454
*PbBES1.7*	229	24,521.34	9.34	53.06	57.16	−0.592
*PbBES1.8*	134	15,094.98	5.72	69.01	73.58	−0.880
*PbBES1.9*	656	73,616.07	5.59	38.65	75.66	−0.450

## Data Availability

The genome sequence data and annotation information of *Phoebe bournei* were downloaded from the Sequence Archive of China National GeneBank Database (CNSA) with accession number CNP0002030.

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
