# Peer review of "Genome-Wide Identification, Characterization, and Expression Analysis of the *BES1* Family Genes under Abiotic Stresses in *Phoebe bournei"

_ijms, 2024, doi:10.3390/ijms25053072_

Round 1

Reviewer 1 Report

Comments and Suggestions for Authors

This article analyzes the physicochemical properties, evolutionary tree, gene structure, conserved structural domains, and intraspecies and interspecies covariance of the BES1 gene family of Phoebe bournei. The expression of the PbBES1 genes in different tissues and different abiotic stress conditions also was determined.

Transcription factors regulate various biological processes by modulating the expression of downstream genes. The tissue-specific expression and abiotic stress response of transcription factors were predictable and preliminary result; thus, this article lacks the substantive innovation. I think that the current manuscript can't be accepted for publication.

However, some suggestions were listed below.

1. Title: deleted the redundant ‘in’.

2. Introduction: This section needs to be condensed and closely followed the theme. The potential relationship between BR and abiotic stress needs to be explained.

3. Why did the authors perform tissue-specific expression analysis of PbBES1?

4. Lines 118-123: contain the grammatical errors.

5. PEG treatment is essentially an osmotic stress that used to simulate drought. It is necessary to pay attention to the wording.

6. Line 153-155: The number of plants used for experiment should be double-checked. 3 plants (Line 153) or 6 plants (Line 155)?

7. Line 350: ‘PbBES1.5 and PbBES1.7 have the 350 highest number of cis-acting elements. This result seems to be inconsistent with the described Figure 6.

8. Figure 10: Working model of BES1 transcription factor under high-temperature stresses was cited from the results of Franklin et al. 2011. This paper is a research article and not a review, so that it is suggested to be deleted.

Comments on the Quality of English Language

English needs to improve.

Author Response

Response to Reviewer 1 Comments:

Dear reviewer,

We feel great thanks for your professional review work on our manuscript. As you are concerned, there are several problems that need to be addressed. According to your nice suggestions, we have made corrections to our previous manuscripts, the detailed corrections are listed below.

Concern (1)  Title: deleted the redundant ‘in’. 

Response(1): We are really sorry for our careless mistakes.Thank you for your reminder.

Concern (2)  Introduction: This section needs to be condensed and closely followed the theme. The potential relationship between BR and abiotic stress needs to be explained.

Response(2): We feel great thanks for your professional review work on our article. As you are concerned, there are several problems that need to be addressed in introduction. According to your nice suggestions, we have made extensive corrections to our previous draft.

Concern (3) Why did the authors perform tissue-specific expression analysis of PbBES1?

Response(3): We sincerely thank you for careful reading. To gain deeper insight into the roles and regulatory mechanisms of PbBES1s in the growth and development of  P. bournei, and to provide a basis for further stress experiments, we conducted an analysis of the expression patterns of the 9 PbBES1 genes in various tissues.

Concern (4)  Lines 118-123: contain the grammatical errors.

Response(4)Thanks for your careful checks.We have revised to this section.

Concern (5)  PEG treatment is essentially an osmotic stress that used to simulate drought. It is necessary to pay attention to the wording.

Response(5):We sincerely thank you for your review and feedback. We have made changes to this section. (Please see line 445-451)

Concern (6)  Line 153-155: The number of plants used for experiment should be double-checked. 3 plants (Line 153) or 6 plants (Line 155)?

Response(6): We were really sorry for our careless mistakes.Thank you for your reminder.In our resubmitted manuscript, the problem is revised.(Please see line 594 and 601)

Concern (7)  Line 350: ‘PbBES1.5 and PbBES1.7 have the 350 highest number of cis-acting elements. This result seems to be inconsistent with the described Figure 6.

Response(7):  Thanks for your careful checks, we were really sorry for our careless mistakes.In our resubmitted manuscript, the problem is revised.(Please see line 351)

Concern (8)  Figure 10: Working model of BES1 transcription factor under high-temperature stresses was cited from the results of Franklin et al. 2011. This paper is a research article and not a review, so that it is suggested to be deleted.

Response(8): Thank you for your advice. When drawing the working model,, we also refer to these two review literatures, which have been revised in the paper. Thank you for your comments again.

Reviewer 2 Report

Comments and Suggestions for Authors

Journal IJMS (ISSN 1422-0067)

Manuscript ID ijms-2869059

Type Article

Title Genome-Wide Identification , Characterization and Expression Analysis of the BES1 Family Genes under Abiotic Stressin in Phoebe bournei

Authors: Jingshu Li , Honggang Sun , Yanhui Wang , Dunjin Fan , Qin Zhu , Jiangyonghao Zhang , Kai Zhong , Hao Yang , Weiyin Chang * , Shijiang Cao *

Section Molecular Genetics and Genomics

Special Issue Genes Function and Mechanism Identification in Plant Stress Resistance 2.0

Dear Editor

I am sending you a review of Manuscript ID ijms-2869059 intended for publishing in the Journal IJMS (ISSN 1422-0067),

The manuscript should be corrected and resubmitted for another review.

Reviewer

l. 17 - 37.

Abstract

1. make a correction, maintaining the appropriate structure of the abstract

l. 38.

Keywords:

2. Eliminate terms appearing in the title of the manuscript.

3. Abbreviations should be fully defined and what they mean.

l. 40 - 60.

Introduction

4. Abbreviations used for the first time should be explained.

5. Explain the mechanism of signaling pathways.

l. 61 - 95.

6. Explain „…which in turn acts on a large number of BR response elements…”

l. 96 - 116.

7. Eliminate mental shortcuts

“…Additionally, Phoebe bournei also has potential medicinal values, including antimicrobial and antitumor activity [27].”

8. When specifying medicinal properties, please:

a. what biological model of the study (cell lines, animal studies, clinical trials),

b. biologically active substance,

c. what cancer,

d. mechanism of action, etc.

l. 117 - 124

9. When specifying the number of genes (it does not provide substantive information to the reader), you should complete, for example:

a. what they encode

b. binary vectors.

l. 125 - 130

10. Reinforce the rationale for undertaking the topic.

11. Formulate a scientific research thesis.

12. State the precise purpose of the research work.

l. 41 - 130

13. I propose to insert thematic subsections.

14. reflect on the order, thematic passages (l. 96 - 116) lack of smooth transition in the text.

15. Please complete the citations of the most recent publications (authors cite in 2021 - 3 publications, in 2022 - 4 publications, in 2023 - 3 publications).

l. 131 - 209

Materials and Methods

16. Please cite publications that confirm the validity of the methods undertaken.

17. Correction of the notation of degrees Celcius “…95 ◦C…”.

l. 210 - 448

Results and discussion

18. Figure 9, please, complete the description of the X axis.

19. Indicate the application of the presented research.

20. Complete the research perspective for the future.

l. 449 - 463.

Conclusions

21. Please, reflect on this fragment. Can it answer the scientific thesis and the purpose of the work?

„…All of these can 461 help us to improve our understanding of BES1. However, we have significant limitations 462 in our studies of BES1, which is a rich area for research….”.

References

l. 48 - 607.

22. Please correct the References in accordance with the guidelines for authors for example position: 1; 2- lowercase and uppercase letters in the title, 4; 6 - Latin names of species are written in italics, e.t.c.

Author Response

Dear reviewer,

We feel great thanks for your professional review work on our manuscript. As you are concerned, there are several problems that need to be addressed. According to your nice suggestions, we have made corrections to our previous manuscripts, the detailed corrections are listed below.

  1. 17 - 37.

Abstract

Concern (1) Make a correction, maintaining the appropriate structure of the abstract.

Response(1)Thanks for your suggestion, we have revised the Abstract.

  1. 38.

Keywords:

Concern (2) Eliminate terms appearing in the title of the manuscript.

Response(2)Thanks for your careful checks.We have revised the keywords.

Concern (3) Abbreviations should be fully defined and what they mean.

Response(3): Thanks for your careful checks.We have revised the keywords.

  1. 40 - 60.

Introduction

Concern (4) Abbreviations used for the first time should be explained.

Response(4): We feel great thanks for your professional review work on our article. As you are concerned, there are several problems that need to be addressed in introduction. According to your nice suggestions, we have made extensive corrections to our previous draft.

Concern (5) Explain the mechanism of signaling pathways.

Response(5): Thanks for your suggestion, We have added the explanation of the mechanism of action in the discussion.

  1. 61 - 95.

Concern (6) Explain „…which in turn acts on a large number of BR response elements…”

Response(6): Thank you for your careful review. After reading this part, we found that there are somes problems with the language description and have revised.

  1. 96 - 116.

Concern (7)  Eliminate mental shortcuts “…Additionally, Phoebe bournei also has potential medicinal values, including antimicrobial and antitumor activity [27].”

Response(7): Thank you for your careful review. After reading this part, we found that there are somes problems with the language description and have revised.

Concern (8) When specifying medicinal properties, please:

  1. what biological model of the study (cell lines, animal studies, clinical trials),
  2. biologically active substance,
  3. what cancer,
  4. mechanism of action, etc.

Response(8): Thank you for your careful review. Because its medicinal properties are not very relevant to this article, we have decided to delete it, thank you for your reminder

  1. 117 - 124

Concern (9) When specifying the number of genes (it does not provide substantive information to the reader), you should complete, for example:

  1. what they encode
  2. binary vectors.

Response(9): Thanks for your suggestion, We have revised in the discussion.

  1. 125 - 130

Concern (10) Reinforce the rationale for undertaking the topic.

Response(10): Thanks for your suggestion, We have revised in the discussion.

Concern (11) Formulate a scientific research thesis.

Response(11): Thanks for your suggestion, We have revised in the discussion.

Concern (12) State the precise purpose of the research work.

Response(12): Thanks for your suggestion, We have revised in the discussion.

  1. 41 - 130

Concern (13) I propose to insert thematic subsections.

Response(13): Thanks for your suggestion, We have revised in the discussion.

Concern (14) reflect on the order, thematic passages (l. 96 - 116) lack of smooth transition in the text.

Response(14): Thanks for your suggestion, We have revised in the discussion.

Concern (15) Please complete the citations of the most recent publications (authors cite in 2021 - 3 publications, in 2022 - 4 publications, in 2023 - 3 publications).

Response(15): Thanks for your suggestion, we have added new references.

  1. 131 - 209

Materials and Methods

Concern (16) Please cite publications that confirm the validity of the methods undertaken.

Response(16): Thank you for your suggestion, we have added the relevant references.

Concern (17) Correction of the notation of degrees Celcius “…95 ◦C…”.

Response(17): We were really sorry for our careless mistakes.Thank you for your reminder.In our resubmitted manuscript, the problem is revised.

  1. 210 - 448

Results and discussion

Concern (18) Figure 9, please, complete the description of the X axis.

Response(18): Thanks for your reminder, we have added the explanation of X axis and Y axis.

Concern (19) Indicate the application of the presented research.

Response(19): Thanks for your suggestions, we added the relevant content in 3.Discussion.

Concern (20) Complete the research perspective for the future.

Response(20): Thanks for your suggestions, we added the relevant content in 3.Discussion.

  1. 449 - 463.

Conclusions

Concern (21) Please, reflect on this fragment. Can it answer the scientific thesis and the purpose of the work?

„…All of these can 461 help us to improve our understanding of BES1. However, we have significant limitations 462 in our studies of BES1, which is a rich area for research….”.

Response(21): Thank you for your review and suggestions. We have revised this  section.

References

  1. 48 - 607.

Concern (22) Please correct the References in accordance with the guidelines for authors for example position: 1; 2- lowercase and uppercase letters in the title, 4; 6 - Latin names of species are written in italics, e.t.c.

Response(22): Thank you for your review and suggestions. We have revised the reference section.

Round 2

Reviewer 1 Report

Comments and Suggestions for Authors

The quality of the revised version has been improved. However, further revisions and standardization are needed for this mannuscript. Some suggestions are as follows. 

1.Title: ‘Genome-Wide’ should be ‘Genome-wide’.

2.Abstract: The first sentence is not end.

3.Keyword: ‘BESA genes’ seems to appear only once throughout the full text.

4.Based on the standardization, some contents needs to be italicized, such as P<0.05, pI, Phoebe bournei (that appears in the title).

5.Figure 7: ‘##1’ should not be listed. This figure needs to be further modified.

6.Abbreviations should be used when Latin names appear again. 

7.PEG treatment is essentially an osmotic stress that used to simulate drought. This problem seems to have not been seriously resolved. Relevant statements need to be carefully considered and modified.

8.There are still many grammar and tense errors, as well as wrong vocabulary (e.g., investigatesd) in the text. Please improve the English writing or submit the manuscript to a professional scientific editing service. 

Comments on the Quality of English Language

Effort needs to be made to improve the English language.

Author Response

Dear reviewer,

We feel great thanks for your professional review work on our manuscript. As you are concerned, there are several problems that need to be addressed. According to your nice suggestions, we have made corrections to our previous manuscripts, the detailed corrections are listed below.

Concern (1) Title: ‘Genome-Wide’ should be ‘Genome-wide’.

Response(1): We are really sorry for our careless mistakes.Thank you for your reminder.

Concern (2) Abstract: The first sentence is not end.

Response(2): Thanks for your careful checks.We have revised to this section.

Concern (3) Keyword: ‘BESA genes’ seems to appear only once throughout the full text.

Response(3): We were really sorry for our careless mistakes.Thank you for your reminder.In our resubmitted manuscript, the problem is revised.

Concern (4)  Based on the standardization, some contents needs to be italicized, such as P<0.05, pI, Phoebe bournei (that appears in the title).

Response(4)We are really sorry for our careless mistakes.Thank you for your reminder.

Concern (5) Figure 7: ‘##1’ should not be listed. This figure needs to be further modified.

Response(5):We were really sorry for our careless mistakes.Thank you for your reminder.In our resubmitted manuscript, the problem is revised.

Concern (6) Abbreviations should be used when Latin names appear again. 

Response(6): We were really sorry for our careless mistakes.Thank you for your reminder.In our resubmitted manuscript, the problem is revised.

Concern (7)  PEG treatment is essentially an osmotic stress that used to simulate drought. This problem seems to have not been seriously resolved. Relevant statements need to be carefully considered and modified.

Response(7): Thank you for your suggestion.We have made changes in the corresponding section and added references.(Please see line 440)

Concern (8)  There are still many grammar and tense errors, as well as wrong vocabulary (e.g., investigatesd) in the text. Please improve the English writing or submit the manuscript to a professional scientific editing service. 

Response(8): Thank you for your suggestion. Regarding language issues in the article, we have opted for the language polishing service provided by MDPI to revise and edit it. All Latin names have also been reviewed and revised.
